# The Role of the Gut Microbiome and Trimethylamine Oxide in Atherosclerosis and Age-Related Disease

**DOI:** 10.3390/ijms24032399

**Published:** 2023-01-25

**Authors:** Racha El Hage, Nada Al-Arawe, Irene Hinterseher

**Affiliations:** 1Department of Vascular Surgery, Universitätsklinikum Ruppin-Brandenburg, Medizinische Hochschule Brandenburg, Fehrbelliner Str. 38, 16816 Neuruppin, Germany; 2Vascular Surgery Clinic, Charité—Universitätsmedizin Berlin, Corporate Member of Freie Universität Berlin and Humboldt-Universität zu Berlin, and Berlin Institute of Health (BIH), 10117 Berlin, Germany; 3Fakultät für Gesundheitswissenschaften Brandenburg, Gemeinsame Fakultät der Universität Potsdam, der Medizinischen Hochschule Brandenburg Theodor Fontane und der Brandenburgischen Technischen Universität Cottbus—Senftenberg, Karl-Liebknecht-Str. 24–25, 14476 Potsdam, Germany

**Keywords:** gut microbiome, atherosclerosis, TMAO, aging, gut dysbiosis, probiotics, short chain fatty acids

## Abstract

The gut microbiome plays a major role in human health, and gut microbial imbalance or dysbiosis is associated with disease development. Modulation in the gut microbiome can be used to treat or prevent different diseases. Gut dysbiosis increases with aging, and it has been associated with the impairment of gut barrier function leading to the leakage of harmful metabolites such as trimethylamine (TMA). TMA is a gut metabolite resulting from dietary amines that originate from animal-based foods. TMA enters the portal circulation and is oxidized by the hepatic enzyme into trimethylamine oxide (TMAO). Increased TMAO levels have been reported in elderly people. High TMAO levels are linked to peripheral artery disease (PAD), endothelial senescence, and vascular aging. Emerging evidence showed the beneficial role of probiotics and prebiotics in the management of several atherogenic risk factors through the remodeling of the gut microbiota, thus leading to a reduction in TMAO levels and atherosclerotic lesions. Despite the promising outcomes in different studies, the definite mechanisms of gut dysbiosis and microbiota-derived TMAO involved in atherosclerosis remain not fully understood. More studies are still required to focus on the molecular mechanisms and precise treatments targeting gut microbiota and leading to atheroprotective effects.

## 1. Introduction

Atherosclerosis is a chronic disease that affects medium and large arteries of the body through a major increase in the lipoproteins of their intimal layer [1]. Risk factors associated with atherosclerosis include diabetes mellitus, hypertension, dyslipidemia, obesity, and smoking [2]. Diabetes mellitus is a major public health problem that has been a leading cause for mortalities worldwide. Diabetes is characterized by elevated levels of blood glucose, which leads over time to serious damage to the heart, blood vessels, eyes, kidneys, and nerves [3]. As a result, diabetes mellitus is associated with accelerated atherosclerosis, leading to vascular lesions that include cardiovascular disease (CVD), coronary artery disease, cerebrovascular disease, and peripheral arterial disease (PAD), with CVD being the major cause of premature death in diabetes. Another consequence of diabetes mellitus is microangiopathy that occurs in the colon, and which has been reported to be more common in diabetics than non-diabetics [4]. Microangiopathy can also occur in the retina, skin (specifically foot skin/diabetic foot ulcer), nerve, kidney, muscle, and heart of diabetic patients and is associated with the thickening of the capillary basement membrane [5]. Other mediators of diabetes that cause vascular complications include dyslipidemia, chronic hyperglycemia, and insulin resistance. Dyslipidemia and chronic inflammation are among the major causes of the development of atherosclerosis, which causes chronic accumulation of lipid-rich plaque in the arteries in diabetic patients [6,7]. Therefore, the regulation of these chronic diseases is crucial. 

The human gut microbiome is a huge microbial community that plays a vital role in human health. With the development in research, the influence of intestinal flora on human diseases has been gradually revealed. Dysbiosis in the gut microbiota (GM) has been reported to have adverse health effects on the human body that lead to a variety of chronic diseases. Regulation of the GM can provide a potential target for the prevention and treatment of disease. The fermentation products of the gut microbiota are by far the most well studied, and they have been described to have a key role in the maintenance of the gut microbial ecology and the modulation of host immunity and metabolic disease [8,9,10,11,12]. The major fermentation products of the GM that result from dietary fibers are short chain fatty acids (SCFAs), with the most abundant metabolites being acetate, propionate, and butyrate. [12,13]. SCFAs can function as a macronutrient energy source and hormone-like signaling molecules that enter the portal circulation to signal through specific host receptor systems in order to regulate the innate immunity and host metabolism. Most studies linking the GM to disease designate SCFAs as potential disease-moderating or prevention factors in metabolic disease, intestinal immunity, cancer, and liver disease [12,13]. Recent reports have indicated that dysbiosis is increased with aging, and that the GM of elderly people is enriched with pro-inflammatory commensals and fewer beneficial microbes [14]. Dysbiosis is presumed to be the primary cause of age-associated morbidities, and, consequently, the premature death of elderly people [14]. Gut dysbiosis leads to a disruption of the microbial metabolites, impaired function of the gastrointestinal tract, and increased leakage of the gut [14]. These events enhance systemic inflammation which is associated with aging, termed inflammaging, and they, consequently, result in aging-associated pathologies [14].

Trimethylamine (TMA) is a byproduct generated from the gut microbial metabolism of dietary amines such as choline, betaine, and carnitine that originate from animal-based foods [15,16]. TMA is absorbed into the portal circulation and is oxidized by the liver into trimethylamine-N-oxide (TMAO) using the flavin monooxygenase enzyme (Figure 1) [16,17]. More attention has been directed upon circulating TMAO due to its pro-inflammatory, pro-atherogenic, and pro-thrombotic properties [18,19,20,21,22]. Several factors, such as diet, gut microbial flora, drug administration, and liver flavin monooxygenase activity, influence the plasma TMAO level [16]. TMAO has been described as vital for lipid balance and for the increase in scavenger receptors, such as CD36 and scavenger receptor class A type 1 (SR-A1), that contribute to the surge of fat accumulation in foam cells, which in turn play a major part in atherosclerotic plaque progress [16,18,23]. In addition, the hepatic enzyme flavin-containing monooxygenase 3 (FMO3) is considered the most active in converting TMA into TMAO, leading to higher plasma TMAO levels. High levels of plasma TMAO have been linked to an alteration of reverse cholesterol transport [16,24], hyperlipidemia and hyperglycemia [16,24,25], and the overexpression of inflammatory markers including tumor necrosis factor alpha (TNF-α), interleukin-6 (IL-6), c-reactive protein (CRP) [26,27], and insulin resistance [16,25], which all lead to the promotion of atherosclerosis [16,19,25,28,29]. It has also been reported that high plasma levels of TMAO metabolite are related to the prognosis of 5-year all-cause mortality in stable patients diagnosed with peripheral artery disease [28,30]. Furthermore, Brunt et al. confirmed that circulating TMAO was high in older compared with younger adults, and that elevated TMAO was correlated with a higher carotid–femoral pulse wave velocity (PWV). Their findings in humans represented the first link between the age-related increase in circulating TMAO with higher aortic stiffness and blood pressure (BP). Nevertheless, as both aortic stiffness and BP are aging risk factors, it might be that both outcomes are in fact not causally associated with TMAO [31].

## 2. TMAO, TMA-Producing Gut Bacteria, and Atherosclerosis

Previous studies described that many human gut colonizing bacteria are capable of producing TMA, which leads to an increase of TMAO levels in plasma. These gut bacteria include *Streptococcus sanguinis, Desulfovibrio alaskensis, Desulfovibrio desulfuricans, Acinetobacter, Serratia, Escherichia coli, Citrobacter, Klebsiella pneumoniae, Providencia, Shigella, Achiomobacter,* and *Sporosorcine*, which belong to the Firmicutes and Actinobacteria phyla. [16,23]. On the other hand, bacteria belonging to phylum Bacteroidetes are not able to produce TMA [16,32]. Several previous studies have investigated the impact of host factors, such as diet and dietary compounds, on TMAO plasma levels, and it was reported that higher plasma TMAO levels have been linked to an animal-based diet [33,34] compared to vegetarians (Figure 1) [19,35]. Mainly, TMA is generated by the enzymes produced by the gut microbiota, and its levels are dependent on the amount of precursors available and the abundance and activity of bacteria catalyzing TMA formation [36], which compete with the host for these precursors that are usually absorbed as essential nutrients [36]. As the quantification of TMA producers in the gut is limited, it is more difficult to understand the distribution of these bacteria in the gut and specify their niches, which is important for designing effective and sustainable treatment strategies to minimize TMAO plasma concentrations [36]. In order to discover the abundance and diversity of TMA-forming gut bacteria, Rath et al. developed assays that targeted key genes encoding enzymes responsible for TMA formation from choline (choline-TMA lyase, *CutC*), carnitine (carnitine monooxygenase, *CntA*), and betaine (subunit B of betaine reductase, *grdH*) [17]. In addition to the association of TMAO with PAD and atherosclerosis, advancing age has also been strongly linked to TMAO levels. As age increases, the host’s physiology and function are altered. For example, the epithelial integrity of the colon, which is needed to promote the influx of bacterial metabolites, including TMAO, might be reduced [37,38]. Rath et al. reported that there was an association between carotid intima-media thickness (IMT) and TMAO only in individuals above 65 years of age, which indicates that aging people are principally affected by this metabolite [38]. Age-related associations between TMAO plasma levels and health parameters have not yet been reported in patients with PAD. Preclinical studies in both animal and in vitro models on human-derived material have emphasized the contribution of TMAO to endothelial senescence and vascular aging [39,40]; however, age-specific effects of TMAO have still not been fully studied. Many studies have investigated the link between diet and TMAO levels, reporting different results; however, an analysis of gut microbiota has only recently been included [38]. Rath et al. were able to provide important information regarding the formation of TMAO in the general population, and they have elaborated on the functional role of the gut microbiota and specific foods, clarifying the increased levels of TMAO with increasing age [38].

## 3. Gut Dysbiosis, Aging, and TMAO Levels

Gut dysbiosis is the disruption in the gut microbiome that is associated with different diseases. Dysbiosis disturbs the gut barrier function, leading to the leakage of harmful metabolites, such as lipopolysaccharides (LPS), and other bacterial components, such as peptidoglycans, into the circulation, which triggers an inflammatory response leading to atherosclerosis (Figure 1) [41,42,43]. LPS can stimulate the uptake of modified low-density lipoprotein (LDL) and reduce the efflux of cholesterol from foam cells, promoting monocyte recruitment and macrophage foam cell formation [41,43,44]. LPS can induce vascular inflammation directly or by producing pro-inflammatory factors from immune cells [44]. The increased production of pro-inflammatory cytokines promotes oxidative stress and oxidized LDL (oxLDL), increasing the risk of hypertension via nitric oxide synthase inhibition. This effect reduces vasodilator nitric oxide levels and increases levels of vasoconstrictor endothelin-1 [41].

### Aging, Gut Dysbiosis, and TMAO

Aging leads to several changes in cells, tissues, and organs [45] and is influenced by an individual’s genetics, lifestyle, and environment [46]. The term “immunoscence” first appeared a few decades ago to refer to impaired or faulty immune responses leading to a decrease in the ability to trigger the immune response and effectively produce antibodies against different pathogens [47,48]. The gut microbiome undergoes dynamic changes through time, and gut dysbiosis is an age-related complication caused by host senescence, changes in nutritional behavior, drug use, and the lifestyle of aged people [48]. The changes in the gut microbiome include shifts in bacterial composition and metabolic function [49]. In humans, age-related gut dysbiosis is characterized by increased inter-individual variation and decreased species diversity; specifically, a loss of Clostridiales and Bifidobacterium, an enrichment of Proteobacteria, Lactobacilli, and an overrepresentation of pathobionts such as Enterobacteriaceae [49,50,51,52]. However, the major gut microbiota aging feature is the decreased ratio of Firmicutes/Bacteroidetes [53]. Schneeberger et al. reported that aged mice showed a decrease in beneficial gut bacteria, such as in Clostridium members of cluster IV that produce SCFAs and *Akkermansia muciniphila*, and an increase in pro-inflammatory microbes [14,54]. Overall, the decrease in intestinal commensal microbes diversity is associated with increased susceptibility to pathogen infection accompanied by disturbance of the gut mucosal barrier and enrichment in pro-inflammatory cytokines; all these events have a detrimental consequence in aging [14,55]. Recent studies suggested that gut dysbiosis is associated with the development of several chronic diseases including cardiovascular disease and other metabolic disorders [56].

Many studies have reported a close relationship between TMAO levels, aging, and age-related diseases. Several animal models have been used to identify mechanisms that underlie TMAO’s role in senescence [57]. Cell senescence involves many processes including increased production of reactive oxygen species (ROS), mitochondrial dysfunction, and senescence-associated secretory phenotype (SASP) [57]. Ke et al. reported that senescence-accelerated prone mouse strain 8 (SAMP8) and senescence-accelerated mouse resistant 1 (SAMR1) were treated with 1.5% (w/v) TMAO for 16 weeks to induce vascular aging and advanced vascular aging processes, respectively [40]. Many potential mechanisms underlie TMAO’s role in aging, including the inhibition of sirtuin 1 (SIRT1) expression, which increases oxidative stress and results in the activation of the p53/p21/Rb pathway. Increased P53 and P21 acetylation and reduced CDK2, cyclinE1, and Rb phosphorylation are followed by enhanced endothelial cell senescence and vascular aging [58]. In addition, TMAO increases the accumulation of ROS, matrix metalloproteinase 2 (MMP2), and matrix metalloproteinase 9 (MMP9) in vivo and in vitro, which are associated to oxidative stress in cells [59]. Furthermore, high TMAO levels are linked to increased expression of pro-inflammatory cytokines, such as TNF-α and IL-1β, as well as decreased production of anti-inflammatory cytokines such as IL-10 [26].

## 4. SCFAs and Their Function in Atherosclerosis

Short chain fatty acids (SCFAs) are the primary fermentation products of dietary fibers and non-digestible carbohydrates (NDCs) [60,61]. NDCs are an important fraction of dietary fibers, and SCFAs are the main products of the favorable saccharolytic fermentation of NDC in the gut [62]. In elderly people, the level of carbohydrate-derived SCFAs is decreased, while the metabolites resulting from protein fermentation, such as phenols, ammonia, and branched fatty acids, are increased. This indicates a shift from saccharolytic fermentation to unfavorable proteolytic fermentation [14,50]. A shift in SCFA production occurs progressively during the aging process, and it is accelerated upon the usage of antibiotics and changes in diet [50].

Several bacterial families, including anaerobic Bacteroides, Bifidobacterium, Eubacterium, Streptococcus, Lactobacillus, clostridial clusters IV and XIVa of Firmicutes, including species of Eubacterium, Roseburia, Faecalibacterium, and Coprococcus, are involved in the production of SCFAs [61,63]. SCFAs mediate the interaction between the gut, diet, and microbiota, highlighting their essential role in intestinal health [61,64,65,66]. The major SCFAs are acetate, propionate, and butyrate, which account for approximately 90% of the total SCFAs formed in the human colon by colonic microorganisms [67,68,69]. They help in the regulation of host metabolic processes to achieve host homeostasis. SCFAs have a high abundance in the gastrointestinal tract (GIT) and are utilized by intestinal epithelial cells (IECs) [70]; they are able to modify several crucial cellular processes including gene expression, proliferation, differentiation, and chemokines production [71]. 

The function of SCFAs is mediated by the activation of six G protein-coupled receptors (GPCR) encoded by the human genome: GPR41 (free fatty acid receptor 3; FFAR3), GPR42, GPR43 (FFAR2), GPR109a (HCAR2), GPR164 (OR51E1), and OR51E2. GPCR41 and GPCR43 are expressed in adipose tissue, intestines, and immune cells [71].

The epithelial cells are in direct contact with high SCFA concentrations, so SCFAs are uptaken into the IEC cytosol via monocarboxylate transporters, such as monocarboxylate transporter-1 (Slc16a1) and the sodium-dependent monocarboxylate transporter-1 (Slc5a8) [72]. Several studies showed that SCFAs improve immune defenses; for instance, butyrate increases the expression of many antimicrobial peptides including LL-37 and CAP-18 [73]. SCFAs are believed to have a universal anti-inflammatory effect by upregulating anti-inflammatory cytokines and downregulating pro-inflammatory cytokines [7,74]. Kim et al. showed that SCFAs enhance the production of cytokines and chemokines, including TNF-α, IL-6, CXCL1, and CXCL10, in ICE in vitro [75]. Butyrate has been demonstrated to reduce atherosclerotic development in animal models via the reduction of pro-inflammatory factors [76]. Aguilar et al. showed in an atherosclerotic mice model with atherosclerosis-prone apolipoprotein E-deleted (ApoE−/−) that consumed a diet containing 1% butyrate for 10 weeks, atherosclerotic lesions in the aorta were reduced by 50%, suggesting a more stable fibrous cap. Moreover, the mice showed a lower macrophage infiltration and elevated collagen deposition; this phenomenon was linked to a decreased CD36 expression in both endothelial cells and macrophages, nuclear factor-κB (NF-κB) activation, and pro-inflammatory cytokines production [76].

The inhibition of histone deacetylases (HDACs), stimulation of histone acetyltransferase, and stabilization of hypoxia-inducible factor (HIF) activity are the second major mechanisms involved in SCFAs’ action [70,77,78,79]. This results in the regulation of gene expression and the inhibition of a vast array of downstream consequences. However, the understanding of SCFA-mediated inhibition of HDACs is still unclear [80]. In this context, SCFAs increase IECs’ oxygen consumption, causing a decrease in oxygen tension and stabilization of HIF [70,81].

Propionate has been reported to act as an anti-atherosclerotic agent via its positive effects on immunity and immune system components, and its ability to reduce plasma lipid levels [82]. Haghikia et al. reported that propionate is able to control cholesterol hemostasis and reduce the aortic atherosclerotic lesion area in ApoE–/– mice fed a high-fat diet (HFD) [83]. In addition, they stated that propionate increases T regulatory (Treg) cell numbers, thus elevating the IL-10 levels in the intestinal wall. IL-10 suppresses the expression of transmembrane transporter Niemann–Pick C1-like 1 (NPC1L1), which is responsible for intestinal cholesterol absorption. Their results were further translated into humans, in which propionate supplementation was able to reduce LDL and TC in hypercholesterolaemic patients [83]. This indicates the potential therapeutic effect of propionate, which modulates the intestinal immune system, thus improving cardiovascular health and preventing atherosclerotic cardiovascular disease [83]. Similarly, Bartolomaeus at al. reported the immunomodulatory effect of propionate, its ability to reduce atherosclerotic lesion in ApoE–/– mice, and the susceptibility to cardiac ventricular arrhythmias of angiotensin II–infused wild-type NMRI mice [84].

The metabolic pathways of both acetate and propionate participate in the regulation of lipid biosynthesis [85]. The ratio of propionate to acetate is a crucial factor in lipid metabolism, in which acetate contributes to lipid synthesis, while propionate reduces fat deposition in the liver and visceral organs [85,86].

## 5. Butyrate-Producing Bacteria and Atherosclerosis

Butyrate has been reported to have an important role in inflammatory diseases in addition to its significant lipid-lowering, anti-oxidant, and insulin resistance-improving effects [7,87,88]. Butyrate also serves as a primary fuel for colonocytes [89]. It acts as a histone deacetylase inhibitor and ligand to GPCRs, affecting cellular signaling in target cells such as enteroendocrine cells [89]. Metagenomics studies reported that diabetes is associated with an altered gut microbiota. An altered gut microbiota leads to a shift in SCFA production. For instance, a lower abundance of butyrate-producing bacteria has been detected in patients with type 2 diabetes [89]. As a result, treatment strategies for diabetes have been developed to increase intestinal levels of butyrate. These strategies involve supplementation with butyrate-producing bacteria, together with dietary fiber, or via fecal microbial transplant from healthy subjects [89]. 

Moreover, studies have provided evidence that butyrate is able to regulate the antioxidant effect of NF-*k*B in endothelial cells and macrophage-mediated lipid metabolism [7]. It is also able to alleviate the production of pro-inflammatory cytokines, such as IL-1β, TNF-α, IL-6 [90], IL-12, and interferon-γ (IF-γ), and it is able to upregulate the production of anti-inflammatory IL-10 by monocytes in vitro [91,92]. Furthermore, butyrate is able to attenuate the release of vascular cell adhesion molecule-1 (VCAM1) and chemotaxis protein-1 (MCP1/CCL2) in vitro, thus reducing the migration and adhesion of monocytes in the lesion area [93]. Increasing evidence shows that butyrate can regulate the occurrence and development of atherosclerosis [7,94]. As such, butyrate is considered a collateral in the prevention and treatment of atherosclerosis [90]. 

Considering the treatment strategies that involve increasing intestinal butyrate levels, next generation probiotics and butyrate-producing bacteria together with prebiotics could be a promising approach. Different next generation probiotics have shown to have a positive effect in diabetes and atherosclerosis.

To start with, *A. muciniphila*, a mucin-degrading gut bacterium, has been inversely linked to diabetes, inflammation, and metabolic disorders [95]. This bacterium has probiotic properties and has been reported to be more abundant in healthy subjects than in patients with diabetes or other metabolic disorders [95,96,97]. The probiotic properties of *A. muciniphila* could be associated with its ability to modulate mucus thickness and gut barrier integrity [95]. Studies using mice have shown that supplementation of *A. muciniphila* resulted in the restoration of mucus thickness that was disrupted in obese and type 2 diabetic mice because of HFD. In addition, this treatment with *A. muciniphila* was able to improve the metabolic profile and reduce the level of serum lipopolysaccharides (LPSs) [95,98]. High levels of serum LPSs have been linked to gut permeability, and, thus, the disruption of intestinal mucus [95,99]. Li et al. reported that *A. muciniphila* reduces atherosclerotic lesions by improving metabolic endotoxemia-induced inflammation through the restoration of the gut barrier [100]. Studies indicate that, despite using mucin as a source of nutrients, *A. muciniphila* is positively linked to mucus thickness and intestinal barrier integrity in both humans and animals [95,98,101]. Moreover, human and animal studies have also shown that *A. muciniphila* is able to improve insulin sensitivity and glucose homeostasis, in addition to modulating obesity by regulating metabolism and energy homeostasis [98,102]. Bodogai et al. reported that the series of inflammatory events that manifest in insulin resistance occur in aged mice as a result of the decrease in abundance of *A. muciniphila* [103]. This has been linked to the reduction of the mucin layer in the colon, which in turn leads to the loss of butyrate-producing commensal bacteria, such as *Intestinimonas butyriciproducens (I. butyriciproducens), Faecalibacterium prausnitzii (F. prausnitzii), Roseburia faecis (R. faecis),* and *Anaerostipes butyraticus (A. butyraticus),* thus leading to an SCFA reduction, specifically of butyrate both in the gut lumen and in the circulation. The reduction in butyrate promotes dysbiosis and leakage from the gut in aged mice, which sustains inflammaging [103]. 

Another butyrate-producing gut bacterium that contributes to different diseases, such as type 2 diabetes, atherosclerosis, antiphospholipid syndrome, and inflammatory bowel disease, is *Roseburia intestinalis. R. intestinalis* has been shown to reduce intestinal inflammation by enhancing the proliferation of Tregs and stimulating the secretion of anti-inflammatory cytokines IL-10, TGF-b, and thymic stromal lymphopoietin (TSLP) [61]. These results proposed a significant immunomodulating and anti-inflammatory effect of the butyrate produced by *R. intestinalis* in the gut [104]. Kasahara et al. reported that *R. intestinalis* was able to reduce endotoxemia, inflammatory markers in plasma and aorta, and the extent of atherosclerotic lesions. These positive effects were attained after the interaction of *R. intestinalis* with dietary plant polysaccharides [105]. 

*F. prausnitzii*, a butyrate-producing gut bacterium, is another next generation probiotic that is dominant in healthy adults and is described for its anti-inflammatory properties and its potential therapeutic effect in patients in Crohn’s disease [13,106,107]. In addition to its beneficial role in bowel disease, *F. prausnitzii* could have a positive effect in obese and diabetic patients due to its ability to produce butyrate. Butyrate has been reported to activate GPCR, thus facilitating the downstream control of gut alterations during obesity and diabetes [108,109].

## 6. Prebiotics in Atherosclerosis

Prebiotics are non-digestible dietary products that can be fermented by the gut microflora and stimulate the growth of beneficial bacteria that colonize the gut. Prebiotics and probiotics are both able to modulate the gut microbiome, resulting in favorable effects for the host. A study by Chen et al. discussed the positive effect of resveratrol (RSV), which is a natural polyphenol with prebiotic benefits, on gut health and, specifically, its ability to cause a reduction in TMAO levels in vivo [110]. RSV naturally occurs in grapes, berries, and other dietary constituents, and is described to be beneficial in the treatment of many metabolic diseases, including atherosclerosis [111]; however, its bioavailability is not high. Evidence elucidated that phenolic phytochemicals with poor bioavailability can act through remodeling the gut microbiota. It has been reported that a polyphenol-rich cranberry extract and metformin were able to reduce diet-induced metabolic syndrome in mice by altering the gut microbiota [112,113]. For instance, studies found that consumption of RSV can modulate the growth of specific gut microbiota in vivo; this included an increase in the Bacteroidetes-to-Firmicutes ratio, and the growth of *Bacteroides, Lactobacillus*, and *Bifidobacterium* [114,115,116,117,118]. As such, RSV was suggested as a potential prebiotic that could promote the growth of beneficial bacteria that confer health benefits to the host. Considering the association between TMAO levels, the gut microbiota, bile acid (BA) metabolism, and atherosclerosis, Chen et al. examined the effect of RSV on TMAO-induced atherosclerosis and the other mentioned factors in C57BL/6J and ApoE/mice. They were able to prove that RSV attenuated TMAO-induced atherosclerosis by decreasing TMAO levels and increasing hepatic BA neosynthesis through a remodeling of the gut microbiota. Moreover, they also showed that RSV-induced BA neosynthesis was partially mediated via downregulation of the enterohepatic farnesoid X receptor–fibroblast growth factor 15 (FXR/FGF15) axis [110].

## 7. Probiotics, TMAO, and Atherosclerotic Lesions

Probiotics are defined as “live strains of strictly selected microorganisms which, when administered in adequate amounts, confer a health benefit on the host’’ [119,120]. To date, conventional probiotics include lactic acid bacteria and some yeasts [13]. The gut microbiota has immunoregulatory functions and can affect the host’s energy harvest [121], in addition to its effect on lipid metabolism [122] and intestinal barrier integrity [123,124]. Different factors indicate the major role of the gut microbiota in the pathogenesis of atherosclerosis. These factors include the major role of gut microbiota in generating atherogenic substances, such as TMA, the shared bacterial phylotype between atherosclerotic plaque and the oral and gut microbiome, and the specific gut metagenome in atherosclerotic patients [125]. It has been reported that some probiotics can support gut barrier functions, thereby reducing the translocation of bacterial and other immunogenic material from the gut [124,126]. Evidence also associated the use of probiotics with a reduction in different cardiovascular disease risk biomarkers, such as serum LDL and total cholesterol (TC), in addition to systemic inflammation [124,127]. Supplements with adequate probiotics were able to improve major atherosclerotic risk factors such as dyslipidemia, hypercholesterolemia, chronic inflammation, and hypertension [128,129]. Huang et al. reported the positive effect of a Lactobacillus strain in the reduction of atherosclerosis lesion area [130]. In addition, several meta-analyses reported a reduction in TC and LDL-C after the intake of probiotics and, specifically, *Lactobacillus acidophilus* [131,132]. Moreover, it has been reported that probiotics are able to improve the integrity of the epithelial barrier and support the function of tight junctions which can inhibit the translocation of harmful metabolites, such as TMAO and LPS, from entering the peripheral circulation leading to a stable atherosclerotic plaque (Figure 1) [61,133]. VSL-3 is a well-studied probiotic mixture containing eight different probiotic strains: *Bifidobacterium breve, Bifidobacterium longum, Bifidobacterium infantis, Lactobacillus acidophilus, Lactobacillus plantarum, Lactobacillus paracasei, Lactobacillus bulgaricus*, and *Streptococcus thermophilus*. Different studies have shown the beneficial effects of VSL-3 in different diseases, including ulcerative colitis [134], liver disease [135], and Crohn’s disease [136]. Moreover, VSL-3 showed a promising potential in the treatment of atherosclerosis. Mencarelli et al. reported that VSL-3 was able to improve insulin signaling and protect against non-alcoholic steatohepatitis and atherosclerosis in ApoE−/− mice with DSS-induced colitis [137]. In addition, Chan et al. compared the effect of VSL-3 with a positive control drug, telmisartan, that has proved to be effective in reducing atherogenesis in ApoE−/− mice [124]. The results showed that VSL-3 was comparable to telmisartan in reducing the biomarkers of vascular inflammation and development of atherosclerosis [138]. The ability of probiotics to support overall gut health has led to more research showing the promising therapeutic effects of probiotics on disease. In fact, probiotics are currently used for the treatment or prevention of irritable bowel syndrome, inflammatory bowel diseases, gluten intolerance, antibiotic-associated diarrhea, and gastroenteritis [139]. Recent evidence has reported the contribution of the gut microbiota in different diseases via the gut–brain axis, gut–liver axis, gut–lung axis, and gut–vascular axis [41,44,126,139]. Furthermore, probiotics can modulate host immune responses [43,140,141], yet the interactions between probiotics, the gut, and the host immune system are very complex and are not fully understood. Studies have reported the positive role of probiotics in inflammation. For example, ApoE−/− mice treated with *Pediococcus acidilactici* R037 showed a reduction in atherosclerotic lesion development via the suppression of pro-inflammatory cytokine production and IF-γ-producing CD4+ T cells [142]. Probiotics are also able to reduce inflammation by increasing the number of Treg cells [143]. A study conducted using VSL-3 showed that the DNA from this consortium was able to limit epithelial pro-inflammatory responses and attenuate the release of TNF-α in response to an *Escherichia coli* DNA injection [144]. Moreover, another study reported the ability of VSL-3 to reduce vascular inflammation in ApoE−/− mice fed a HFD [124]. It has also been shown that the DNA from VSL-3 was able to employ anti-inflammatory effects through TLR9 signaling [145]; the authors also reported that the protective anti-inflammatory effect of probiotics was reconciled through their DNA and not through their metabolites, and that signaling of TLR9 had a major role in mediating this effect [43,146]. 

Nonetheless, not all probiotics have a proactive role in the treatment of atherosclerosis. For instance, *Lactobacillus reuteri* showed no effect on atherosclerosis in ApoE−/− mice fed a HFD [43,147]. Huang et al. examined the effects of two Lactobacillus strains (*L. acidophilus* ATCC 4356 and 4962) on atherosclerosis development and atherosclerotic lesions in ApoE−/− mice [130]. They reported a dramatic reduction in the atherosclerotic lesion area in the L.4356 group; however, no significant effect was observed in the L.4962 group [130]. In addition, L.4356 was able to significantly reduce plasma cholesterol levels [130]. In another study, Chen et al. reported that L.4356 was able to attenuate the atherosclerotic lesion development in ApoE−/– mice through reducing inflammatory response and oxidative stress [148]. Moreover, Qiu et al. investigated the potential TMAO lowering property of five different probiotics strains, and only *Lactobacillus plantarum* ZDY04 was able to significantly lower the plasma TMAO levels. This was achieved through the remodeling of the gut microbiota, and not by affecting the expression of hepatic FMO3 and metabolizing choline, TMA, and TMAO [149]. Similarly, another study reported the TMAO-lowering potential of *Enterobacter aerogenes* ZDY01 in choline-fed mice; the effect was also attained through gut remodeling [150]. The TMAO-lowering property is strain specific, as a human study investigating the supplementation of *Streptococcus thermophilus* (KB19), *Lactobacillus acidophilus* (KB27), and *Bifidobacteria longum* (KB31) stated that there was no effect on plasma TMAO levels [151]. Another study by Tripolt et al. reported no effect on TMAO levels after 12 weeks of supplementation with *Lactobacillus casei Shirota* in patients with metabolic syndrome [152]. The mechanisms underlying the effects of probiotics on host health are still not fully understood [61]. As a result, the use of probiotics that can directly act on the TMA in the gut might be an alternative approach to reduce serum TMAO levels and prevent the development of atherosclerosis and “fish odor syndrome” [149].

## 8. Mechanisms Underlying the Therapeutic Effect of Probiotics in Atherosclerosis

The mechanisms underlying the protective effect of probiotics against atherosclerosis are not fully understood. Nevertheless, the action of probiotics at different steps is becoming clear as more studies are being conducted. To start with, it has been reported in previous studies how probiotics can combat gut dysbiosis through strengthening the epithelial tight junctions, preventing the translocation of damaging metabolites, such as LPS and TMAO, into the circulation, which can lead to a stable atherosclerotic plaque [61]. Several studies have demonstrated the hypocholesterolemic effect of probiotics [153,154]. Liong and Shah have pointed out the ability of Lactobacillus strains to reduce cholesterol in an in vitro model, and this was achieved through various mechanisms such as assimilation of cholesterol during growth, incorporation of cholesterol into the membrane of cells, the binding of cholesterol to the cell surface, and co-precipitation with deconjugated bile [155]. These findings were backed up in another study by Zeng et al., who studied *Lactobacillus buchneri* P2 and confirmed the cholesterol removal trait of this bacteria through an assimilation mechanism [156]. Another study by Huang and Zheng reported the cholesterol lowering property of a probiotic strain through the inhibition of the gene expression of NPC1L1 in Caco-2 cells [157,158]. The NPC1L1 protein plays a major role in cholesterol absorption, and it is considered to be a promising target for cholesterol-lowering medication [159]. NPC1L1 has been identified by Duval et al. to be a novel target gene for the liver X receptors (LXRs), which support the crucial role of LXRs in intestinal cholesterol homeostasis [160]. LXRs activation has been reported to reduce whole-body cholesterol levels and reduce atherosclerosis [161]. In addition, VSL-3 was found to improve lipid profiles in mice [162], and this was attained by promoting BA deconjugation and fecal excretion, and by increasing hepatic BA synthesis through the downregulation of the FXR/FGF15 axis [163]. BAs can regulate cholesterol balance, and disruption in the circulation of enterohepatic BAs can lead to gall bladder [164] and gastrointestinal diseases [165]. The metabolism of BAs is also associated with obesity, diabetes, and cardiovascular diseases [166,167,168]. BAs are synthesized from hepatic cholesterol, and they are further conjugated with amino acids glycine and taurine to form bile salts that are transferred to the intestine. Bile salts’ amphiphilic combination is crucial for fat absorption in the intestine, yet excessive bile salts are toxic to the gut bacteria [169]. Bile salt hydrolase (BSH), which is present in the gut microbiome, is responsible for the catalysis of the conjugated bile salts into deconjugated BAs in order to maintain the balance of metabolism of BAs. Deconjugated BAs function as signaling molecules to aid in the secretion of GLP-1 hormone [170], activate other receptors, and impact different metabolic processes involved in various diseases [171,172]. The presence of BSH has been identified in different microbial genera such as *Lactobacillus, Bifidobacterium, Enterococcus, Clostridium spp.,* and *Bacteroides.* It has also been reported that one bacterial strain can possess distinct BSHs that can have different properties [173]. In addition, recent evidence suggested that TMAO can promote atherosclerosis, partially through inhibiting hepatic bile acid synthesis [110]. 

In addition, probiotics can apply their anti-inflammatory actions through modulating the expression of key transcription factors or microRNAs (miRNAs), which are associated with pro-inflammatory signaling [148,174]. For instance, Chen et al. reported that *L. acidophilus* ATCC 4356 was able to reduce the levels of TNF-α and oxidative stress markers in addition to its ability of reversing the reduction in IL-10 levels via inhibiting the activation of NF-κB and its translocation to the nucleus [148]. In the case of atherosclerosis, T lymphocytes and macrophages accumulate and proliferate at the atherosclerotic lesions, which leads to the secretion of inflammatory cytokines such as TNF-α and IL-10 [148]. These inflammatory cytokines can, in turn, activate intracellular NF-κB signaling pathways which can stimulate the production of more cytokines, leading to further inflammation [175]. Activated NF-κB exists in the fibrotic thickened intima-media and atheromatous areas of the atherosclerotic lesion, smooth muscle cells, macrophages, and endothelial cells; however, little or no activated NF-κB can be detected in vessels lacking atherosclerosis [176,177]. The activation of NF-κB is associated with the phosphorylation of IκB-α and subsequent degradation of IκB-α, which results in the translocation of NF-κB into the nucleus. It has also been reported that TNF-α is one of the inflammatory markers that can promote atherosclerosis. On the other hand, IL-10 is an anti-inflammatory marker that provides a crucial atheroprotective signal [44]. As for the miRNAs, *Lactobacillus acidophilus* has been shown to protect against apoptosis and necrosis in human endothelial cells, which is induced by LPS stimulation, and this in turn was associated with a decrease in the expression of pro-inflammatory miR-155 and increased expression of anti-apoptotic mIR-21 [43,174]. Nonetheless, further research is still needed to explain the effect of probiotics on the expression of miRNAs associated with atherosclerosis [43]. More studies showing the atheroprotective effect of different probiotics are described in Table 1.

## 9. Different Strategies Involved in Inhibiting TMAO Formation

Considering the composite nature of TMAO formation, different strategies have been predicted to affect the different pathways for TMAO production and reduce the risk of atherosclerosis and cardiovascular disease. Based on this context, a dietary intervention, such as limiting animal-based foods that leads to TMA, could be a straightforward solution; however, this could lead to clinical consequences due to the deficiency of major nutrients that are required for optimal health [194]. Another research interest has focused on identifying the specific enzymes responsible for TMAO production. FMO3 is an enzyme in the liver that is reported to convert TMA into TMAO, and the inhibition of this enzyme would lead to the accumulation of TMA in the gut, which would cause trimethylaminuria, also known as fish malodor syndrome [195,196]. Therefore, the optimal approach for TMAO reduction would be to decrease TMA formation by the gut microbes [197]. Research described the use of 3,3-dimethyl-1-butanol (DMB), the structural analog of choline, as a drug to reduce TMA formation via the inhibition of microbial choline TMA lyase [23]. The results showed that the use of DMB was able to reduce TMAO circulating levels, in addition to its ability to suppress macrophage foam cell formation and ameliorate atherosclerotic lesion formation in vivo [23]. Moreover, DMB can reduce platelet activation and the thrombus formation rate [197]. Despite all these promising results, the effects of DMB were reversible, meaning a direct injection of TMAO could completely reverse the positive outcomes of DMB. Other choline analogues and second-generation TMA lyase inhibitors include fluoromethylcholine (FMC) and iodomethylcholine (IMC). Both FMC and IMC were able to promote the irreversible inhibition of microbial TMA lyase. They were also able to suppress TMA and TMAO levels, and reduce thrombus formation without any noticed toxicity in vivo compared to DMB [197]. Another agent used was the pharmacological product meldonium, which acts as an analogue of γ-butyrobetaine. Evidence has shown that long-term administration of meldonium decreased the levels of circulating L-carnitine in healthy non-vegetarian individuals by suppressing γ-butyrobetaine hydroxylase enzyme [198]. It has also been reported that treatment with meldonium led to a decreased plasma TMAO concentration through increased urinary excretion [199,200]. These different reported treatments could be promising in the prevention of atherosclerosis or cardiovascular disease [201]. Other studies considered that modulating the gut microbiota composition and metabolic function could be an optimal strategy for reducing TMAO levels, as the gut microbiota has been reported to be a major factor in determining the amount of TMA generated. Some literature has demonstrated that antibiotic administration could decrease levels of TMA-generating bacteria, thus decreasing TMAO levels. However, short-term changes in TMAO levels and the high risk of antibiotic resistance are one of the reasons to disregard the use of antibiotics as a therapeutic strategy to modulate TMAO levels. Another studied strategy for modulating the gut microbiota and its metabolic function was repopulating the gut with other microorganisms that are able to cause a decrease in circulating TMA levels. As such, fecal microbial transplant (FMT) has been reported to successfully transmit atherosclerosis susceptibility [202], in addition to increasing the thrombosis potential [203] and leading to higher platelet reactivity in animal models. On the contrary, in a double-blinded controlled pilot study, the FMT from a lean vegan donor did not improve TMAO levels, and it did not have any effect on the parameters of vascular inflammation, despite it causing changes in the intestinal microbiota composition [204]. Furthermore, Kajllmo et al. investigated the effect of FMT from healthy donors used to treat patients with irritable bowel syndrome (IBS) on plasma lipids and LDL/HDL subfractions in a randomized, double-blinded study; they reported no significant effect of FMT on LDL, HDL, and TC levels [205]. However, a recent study by Kim et al. examined the effect of the gut microbiota on the pathogenesis of atherosclerosis in a transgenic atherosclerosis model with C1q/TNF-related protein 9-knockout (CTRP9-KO) mice. CTRP9 plays an important role in cardiovascular homeostasis, promotes endothelial cell function, and improves endothelial-dependent vasorelaxation. They were able to demonstrate that an FMT from wild-type (WT) mice into CTRP9-KO mice was able to decrease atherosclerotic lesions in carotid arteries. In contrast, wild-type (WT) mice transplanted with FMT from CTRP9-KO mice showed progression of atherosclerosis [206]. These results are promising, but more studies are still needed to prove that FMT is a potential treatment for atherosclerosis or CVD [194,207]. Another repopulating strategy to alter the microbiota that has been used is the administration of probiotics. The effects of probiotics on TMAO circulating levels have been reported previously in our review.

## 10. Conclusions

Recently, gut microbiota and microbial metabolites, including TMA and SCFAs, have attracted the focus of researchers due to their crucial role in the development of atherosclerosis and cardiovascular disease. Gut microbiota composition alters between different age groups, and this compositional shift is associated with immune dysregulation and the onset of aging-associated pathologies such as PAD. Gut dysbiosis in the elderly has been associated with the impairment of intestinal barrier integrity, an increase in gut leakiness, endotoxemia, and subsequent inflammaging. Studies have provided evidence of the beneficial role of probiotics, prebiotics, and SCFAs in the management of several atherogenic risk factors. However, some atheroprotective effects of probiotics were strain-specific, and further research needs to be performed to better understand the mechanisms behind the different effects. Further inspection is also required to confirm the atheroprotective effect of SCFAs, and whether the effect of their supplementation persists over the long term. In addition, targeting TMA and TMAO might act as a potential novel therapeutic strategy to prevent atherosclerosis development, plaque rupture, and cardiovascular disease. Although findings in this regard were promising, the exact mechanisms of gut dysbiosis and microbiota-derived TMAO involved in atherosclerosis are not yet fully understood. More well-conducted studies focusing on molecular mechanisms and precise treatments targeting gut microbiota-dependent metabolites for anti-atherosclerosis and, specifically, in the elderly, remain to be completed in further investigations. Additional studies are also required to better understand the factors leading to TMA-forming bacteria and their consequent therapeutic manipulations.

## Figures and Tables

**Figure 1 ijms-24-02399-f001:**
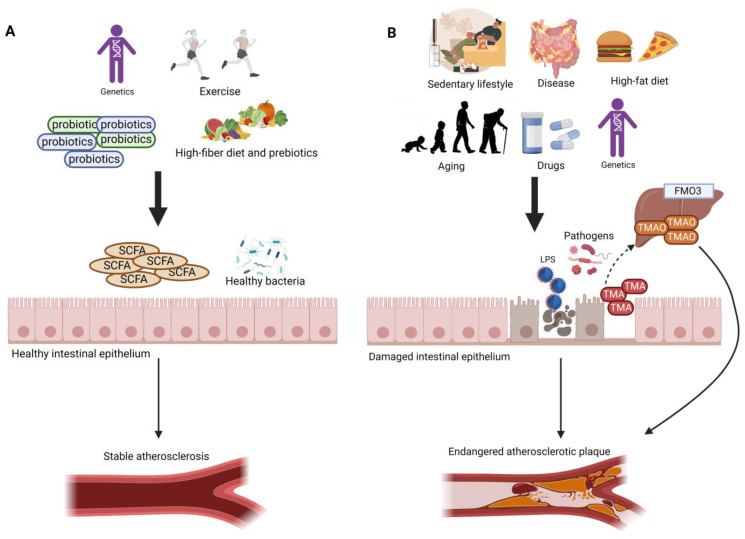
Different factors, such as the gut microbiome, diet, lifestyle, and genetics, play a major role in the development of atherosclerosis. (**A**) High-fiber diet, active lifestyle, and the intake of probiotics increase the abundance of beneficial gut bacteria and the amount of SCFAs, which support the tight junctions and protect the intestinal epithelium, thus preventing harmful metabolites from entering the circulation. This contributes to a stable atherosclerotic plaque. (**B**) High-fat diet, sedentary lifestyle, intestinal disease (e.g., Crohn’s disease, irritable bowel disease), the intake of different drugs (e.g., antibiotics), and aging lead to a disruption in the gut microbial profile, resulting in higher abundance of pathogenic bacteria and lower amounts of SCFAs. These effects lead to damage in the gut epithelium; thus, a leaky gut allows the translocation of LPS, TMA, and other damaging metabolites into the circulation. High amounts of plasma LPS and TMAO result in an endangered atherosclerotic plaque.

**Table 1 ijms-24-02399-t001:** Anti-atherogenic effects of probiotics in human and animal studies.

Reference	Study Design	Study Group	Bacteria Administered	Athero-Protective Outcomes
Chan et al. 2016 [124]	ApoE−/− mice were fed HFD alone or with VSL#3 or a positive control treatment, telmisartan or both for 12 weeks	ApoE−/− mice	VSL#3	VSL#3 reduced pro-inflammatory adhesion molecules and risk factors of plaque rupture, and reduced vascular inflammation and atherosclerosis to a similar extent to telmisartan. Combining both VSL#3 and telmisartan showed no further benefits
Huang et al. 2014 [130]	Eight week-old ApoE−/− mice fed a Western diet with or without *L. acidophilus ATCC 4356* daily for 16 weeks	ApoE−/− mice	*Lactobacillus acidophilus (ATCC 4356)*	*L. acidophilus ATCC 4356* protected ApoE−/− mice from atherosclerosis by reducing their plasma cholesterol levels.
Chen et al. 2013 [148]	Eight week-old ApoE−/− mice treated with *L. acidophilus ATCC 4356* daily for 12 weeks. Wild-type (WT) mice or ApoE−/− mice (control group treated with saline only). Body weight, serum lipid levels, aortic atherosclerotic lesions, and inflammatory status were examined	ApoE−/− mice	*Lactobacillus acidophilus ATCC 4356*	Decreased atherosclerotic lesion size, decreased levels of serum malondialdehyde (MDA), oxLDL, and TNF-α; increased levels of IL-10 and superoxide dismutase (SOD) activity in serum.
Qiu et al. 2018 [149]	Five probiotic strains were investigated for choline-induced TMAO levels in ApoE−/− mice supplemented with 1.3% choline. Only *Lactobacillus plantarum ZDY04 (PLA04)* was subjected for further investigation.	ApoE−/− mice	*Lactobacillus plantarum ZDY01 (PLA01), Lactobacillus rhamnosus ZDY9 (LGG), Lactobacillus plantarum ZDY04 (PLA04), Lactobacillus caseii ZDY8 (CAS), Lactobacillus bulgaricus ZDY5 (BUL)*	*L. plantarum ZDY04* reduced serum TMAO levels and cecal TMA levels, and inhibited atherosclerotic lesion formation. *L. plantarum ZDY04* had no effect on hepatic FMO3 activity.
Qiu et al. 2017 [150]	*Enterobacter aerogenes ZDY01* was administered to choline-fed mice. Serum TMAO and cecal TMA levels were measured	Mice	*Enterobacter aerogenes* ZDY01	Reduction in serum TMAO and cecal TMA levels.
Borges et al. 2019 [151]	21 patients with chronic kidney disease in a double-blind pilot study (3 months duration). A total of 10 patients in the placebo group and 11 patients in the probiotic group. Plasma TMAO, choline, and betaine were measured	Human	*Streptococcus thermophilus* (KB19), *Lactobacillus acidophilus* (KB27), *Bifidobacteria longum* (KB31)	No change in TMAO levels and significant increase in betaine plasma levels after probiotic supplementation. Significant decrease in choline plasma levels in placebo group.
Jones et al. 2012 [178]	114 subjects in a double-blind, placebo-controlled, randomized study received either yogurts containing microencapsulated *L. reuteri* NCIMB 30242 or placebo yogurts	Human	*Lactobacillus reuteri NCIMB 30242*	Reduction in LDL-C, TC, apoB-100, and non-HDL-C.
Rajkumar et al. 2014 [179]	Subjects randomized into four groups: placebo, omega-3 fatty acid, probiotic VSL#3, or both omega-3 and probiotic, for 6 weeks. Blood and fecal samples examined at baseline and after 6 weeks	Human	VSL#3	Reduction in TC, triglyceride(TG), LDL, and VLDL; increased HDL levels, improved insulin sensitivity and decreased hsCRP.
Rerksuppaphol et al. 2015 [180]	Patients diagnosed with hypercholesterolemia received probiotic capsule of *Lactobacillus acidophilus* plus *Bifidobacterium bifidum* three times daily for six weeks. TC, HDL-C, LDL-C, and TG levels were measured	Human	*Lactobacillus acidophilus, Bifidobacterium bifidum*	Decreased TC, HDL-C, and LDL-C levels in probiotic group.
Boutagy et al. 2015 [181]	Nineteen healthy, non-obese males (18–30 years) were randomized to either VSL#3 or placebo during the consumption of a hypercaloric, high-fat diet for 4 weeks. Plasma TMAO, L-carnitine, choline, and betaine (UPLC-MS/MS) were measured at baseline and following a HFD.	Human	VSL#3	Increased plasma TMAO in both the VSL#3 and placebo groups. Plasma L-carnitine, choline, and betaine concentrations did not increase following the HFD in either group. VSL#3 treatment did not influence plasma TMAO concentrations.
Bjerg et al. 2015 [182]	Fecal samples were collected at baseline, after four weeks supplementation, and two weeks after the supplementation was ended; fasting blood samples were collected at baseline and after 4 weeks.	Human	*Lactobacillus paracasei subsp. paracasei, Lactobacillus casei W8®*	Reduced TG
Bernini et al. 2016 [183]	Fifty-one patients with MetS were divided into a control group and a probiotic group. The probiotic group received fermented milk with probiotics for 45 d. The effects of *B. lactis* on lipid profile, glucose metabolism, and pro-inflammatory cytokines were assessed in blood samples.	Human	*Bifidobacterium lactis HN019*	Reduction in body mass index (BMI), TC, LDL-C, TNF-α and IL-6.
Madjd et al. 2016 [184]	Overweight and obese women consumed either a probiotic yogurt (PY) or a standard low-fat yogurt (LF) every day with their main meals for 12 weeks while following a weight-loss program	Human	*Streptococcus thermophiles*, *Lactobacillus bulgaricus*, *Lactobacillus acidophilus* LA5, *Bifidobacterium lactis* BB12	Reduction in TC, LDL-C, insulin resistance, postprandial blood glucose, and fasting insulin.
Chan et al. 2016 [185]	12 weeks feeding of HFD as opposed to normal chow diet (ND) in ApoE−/− mice. LGG or TLM supplementation to HFD was studied	ApoE−/− mice	*L. rhamnosus GG (LGG)*	Reduced lesion development; decreased plasma cholesterol, sE-selectin, sICAM-1, sVCAM-1, and endotoxin.
Costabile et al. 2017 [186]	Double-blind, placebo-controlled, randomized design in which subjects received encapsulated *Lactobacillus plantarum ECGC 13110402* twice daily.	Human	*Lactobacillus plantarum* ECGC 13110402	Reduction in LDL. Reduction in systolic blood pressure.
Firouzi et al. 2017 [187]	A randomized, double-blind, parallel-group, controlled clinical trial included 136 participants with type 2 diabetes, aged 30–70 years who received either probiotics or placebo for 12 weeks.	Human	*Lactobacillus acidophilus, Lactobacillus casei, Lactobacillus lactis, Bifdobacterium bifdum, Bifdobacterium longum,* and *Bifdobacterium infantis*	Improved HbA1c and fasting insulin.
Yoshida et al. 2018 [188]	Oral gavage of *Bacteroides vulgatus* and *Bacteroides dorei* in 6-week-old female ApoE−/− mice 5 times per week for 10 weeks. At 16 weeks of age, the mice were euthanized and analyses were performed to evaluate atherosclerosis.	ApoE−/− mice	*Bacteroides vulgatus, Bacteroides dorei*	Reduced plaque inflammation, attenuating atherosclerotic lesion form.
Saika et al. 2018 [189]	Wistar rats fed a high-cholesterol diet received *Saccharomyces cerevisiae ARDMC1*	Wistar rats	*Saccharomyces cerevisiae ARDMC1*	Reduced TC, LDL, and TG.
Huang et al. 2018 [190]	Oral administeration with *Enterococcus faecium* to rats for 35 days. The gene transcriptions related to cholesterol metabolism, composition of bile acids in feces, synthesis of TMAO in the liver, and composition of the gut microbiota of rats were examined.	Rats	*Enterococcus faecium* WEFA23	Reduction of cholesterol, upregulation of genes’ transcript level relevant to cholesterol decomposition and transportation, and downregulation of genes involved in cholesterol synthesis. Decreased TMAO production followed by increasing the CYP7A1 transcript level.
Szulinska et al. 2018 [191]	81 obese Caucasian women randomly assigned to three groups: a placebo, low dose (LD), and high dose (HD) of lyophilisate powder containing live multispecies probiotic bacteria. The probiotic supplement was administered daily for 12 weeks.	Human	*Bifidobacterium bifidum* W23, *Bifidobacterium lactis* W51, *Bifidobacterium lactis* W52, *Lactobacillus acidophilus* W37, *Lactobacillus brevis* W63, *Lactobacillus casei* W56, *Lactobacillus salivarius* W24, *Lactococcus lactis* W19, *Lactococcus lactis* W58	HD decreased systolic blood pressure, vascular endothelial growth factor, pulse wave analysis systolic pressure, pulse wave analysis pulse pressure, pulse wave analysis augmentation index, pulse wave velocity, IL-6, TNF-α, and thrombomodulin. LD decreased the systolic blood pressure and IL-6 levels.
Tang et al. 2021 [192]	*E.aerogenes* ZDY01 was administered to ApoE−/− mice fed with 1.3% choline	ApoE−/− mice	*Enterobacter aerogenes* ZDY01	Inhibition of choline-induced atherosclerosis. Reduction of cecal TMA and serum TMAO levels and modulation of CDCA-FXR/FGF15 axis.
Wang et al. 2022 [193]	Eight strains of *Bifidobacterium breve* and eight strains of *Bifidobacterium longum* were administered to choline-fed C57BL/6J mice for 6 weeks	C57BL/6J Mice	Eight strains of *Bifidobacterium breve* and eight strains of *Bifidobacterium longum*	*B. breve Bb4* and *B. longum BL1* and BL7 significantly reduced plasma TMAO and plasma and cecal TMA concentrations.

## Data Availability

Not applicable.

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
