# Peer review of "The Role of the Gut Microbiome and Trimethylamine Oxide in Atherosclerosis and Age-Related Disease"

_ijms, 2023, doi:10.3390/ijms24032399_

Round 1

Reviewer 1 Report

The authors aimed  to review the role of the gut microbiome in atherosclerosis and age-related disease. The paper contains an interesting topic. This reviewer suggest to accept the manuscript in current form.

Author Response

The authors would like to thank the reviewer for the feedback and for taking the time to review our manuscript.

Reviewer 2 Report

The manuscript is very well organized and updated and it focuses on TMA and TMAO and on mechanistic insights of disease progression. I think that it is a good review article.The Figure is very representative and the Table is a good guide to find the original articles cited by the review. I would recommend publication of the manuscript.

El Hage and colleagues review the importance of gut microbiome, and specifically of harmful metabolites such as trimethylamine (TMA). TMA is metabolized in the liver and gives rise to trimethylamine oxide (TMAO), which plays a role in atherosclerosis and age-related disease.

Several reviews have been written in the last years about gut microbiome and cardiovascular disease, since it is a hot topic (e.g. https://pubmed.ncbi.nlm.nih.gov/36439214/), but many of them focus on specific subtopics (e.g. https://www.frontiersin.org/articles/10.3389/fcvm.2021.668532/full, https://pubmed.ncbi.nlm.nih.gov/35795187/, https://pubmed.ncbi.nlm.nih.gov/35474231/). For this reason, I think that it would be important to highlight in the title of the manuscript that this review has a focus on TMA and TMAO.

Author Response

The authors would like to thank the reviewer for the feedback and for taking the time to review our manuscript. The authors have taken the reviewer's valuable remark into consideration and have changed the title of the manuscript to be more specific and to indicate the focus of the review. The updated title is: "The role of the gut microbiome and trimethylamine oxide in atherosclerosis and age-related disease".

Reviewer 3 Report

Review by Hage et al., is well-written and summarizes old to recent literature in the present review. I had the suggestions.  

Line 277- typo NFjb I guess its NFkB

please include references at the end of sentences of line number- 69,72,96 and others where needed.

Additionally, I would recommend modifying figure 1. I would start figure 1 with the factors  (diet, microbiome, genetics) that play role in atherosclerosis first (on top) and then show damage in the epithelial barrier.    References are up to date.

Author Response

The authors would like to thank the reviewer for the feedback and for taking the time to review our manuscript. The authors have taken the reviewer's valuable suggestion into consideration and have adjusted the figure accordingly. The authors have moved all the factors contributing to atherosclerosis to the top, and showed the stable atherosclerotic plaque first on the left and then the endangered atherosclerotic plaque on the right side of the figure. The figure title has also been adjusted accordingly. The adjusted figure is uploaded in the revised manuscript version with the track changes. In addition, the typo has been adjusted, and the required references have been added.